# Diurnal Profiles of Locomotive and Household Activities Using an Accelerometer in Community-Dwelling Older Adults with Musculoskeletal Disorders: A Cross-Sectional Survey

**DOI:** 10.3390/ijerph17155337

**Published:** 2020-07-24

**Authors:** Harutoshi Sakakima, Seiya Takada, Kosuke Norimatsu, Shotaro Otsuka, Kazuki Nakanishi, Akira Tani

**Affiliations:** 1Department of Physical Therapy, School of Health Sciences, Faculty of Medicine, Kagoshima University, 8-35-1, Sakuragaoka, Kagoshima 890-8544, Japan; k6745961@kadai.jp (K.N.); k9378361@kadai.jp (K.N.); k4718950017@gmail.com (A.T.); 2Department of Systems Biology in Thromboregulation, Kagoshima University Graduate School of Medical and Dental Science, Kagoshima 890-8520, Japan; karaagetantou0110@gmail.com (S.T.); k3360022@kadai.jp (S.O.)

**Keywords:** physical activity, aging, rehabilitation, musculoskeletal disease

## Abstract

The present study investigates the diurnal profiles of locomotive and household activities in older adults with musculoskeletal disorders (MSDs) using an accelerometer. Furthermore, we examined the effect of chronic pain on their diurnal profiles in both activities. Seventy-one older adults with MSDs (73–89 years) were included in this cross-sectional survey, and 25 age-matched older adults (75–86 years) were selected as healthy older adults. The daily physical activities, including steps walked and locomotive and household activity intensities, were recorded using a triaxial accelerometer in terms of metabolic equivalent task-hours per week (MET-h/week). The diurnal profiles of steps and locomotive activities in older adults with MSDs were considerably lower than those of healthy older adults. In contrast, there was no significant decline in household activity. However, the locomotive and household activities were reduced by severe chronic pain. This survey demonstrated that the diurnal profiles of household activity in older people with MSDs as well as those in age-matched healthy older adults were maintained. Furthermore, severe chronic pain influenced both activities. Therefore, the maintenance of household activity throughout the day, as well as the management of chronic pain, may be important strategies for the promotion of physical activity in older people with MSDs.

## 1. Introduction

Musculoskeletal disorders (MSDs), including chronic musculoskeletal pain and osteoarthritis of the knee or hip, are common in older people. These diseases have a significant impact on locomotive and household activities [1,2], and place a considerable burden on the health care system [3]. Although orthopedic surgery can be performed as a treatment in older people with MSDs, the chronic musculoskeletal pain and physical function may not fully improve after surgery. Therefore, these patients have to live with pharmacological and non-pharmacological therapies, such as ambulatory rehabilitation. In addition, older people with MSDs are not physically active, and they likely do not meet the recommended physical activity levels in daily life [2,4]. Therefore, some community-dwelling older adults with MSDs have preventive care programs to enhance physical function. It is naturally important to enhance daily physical activities, including locomotive and household activities to prevent the requirement of a care program. However, few studies have investigated the diurnal profiles of the physical activities classified as locomotive and household activities throughout the day in community-dwelling older adults with MSDs.

Some studies have reported the diurnal profiles of physical activity in community-dwelling older adults with chronic diseases and functional limitations [5,6]. These studies focused on steps taken throughout the day and health status in older adults. Mai et al. [6] reported that the high activity phases throughout the day were mostly due to basic and instrumental activities of daily living, such as housework. However, they did not actually investigate the diurnal profiles that categorized locomotive or household activities in older adults. Therefore, it is unclear whether the older adults performed housework activities in high activity phases throughout the day. Furthermore, although musculoskeletal pain is a common source of serious long-term pain and physical disability [1], few studies have investigated the effect of the chronic pain level on the diurnal profile of locomotive or household activities throughout the day in older adults with MSDs.

Accelerometers are increasingly being used to allow researchers to assess energy costs, and are generally considered superior to other methods of measuring physical activity categories [7,8]. Accelerometers are non-invasive tools for measuring physical activity and activity energy expenditures, with the potential to measure locomotive as well as household activities [8,9,10]. The Omron activity monitor (Active style Pro, HJA-750C, Omron Healthcare Co., Ltd., Kyoto, Japan), which is a triaxial accelerometer, is a type of motion sensor that is low-cost unobtrusive. Physical activity intensities can be used to estimate the metabolic equivalents (METs) for daily activities from the original algorithm, using the Omron activity monitor [9,11]. A change in the movement of the body was estimated using Omron’s original signal processing, and physical activities were classified into either locomotive or household activities, using the original algorithm [7,9,11,12]. Furthermore, Park et al. [13] examined the association of locomotive and non-locomotive physical activities, such as household activity, using an Omron activity monitor in healthy older men. Many investigators have assessed the validity and reliability of the Omron activity monitor in large populations [9,11,12]. Therefore, we examined the diurnal profiles of locomotive or household physical activities during the course of the day between older adults with MSDs and age-matched healthy older adults. An explorative analysis of locomotive or household activities may provide interesting insights regarding the physical activity patterns of older adults with MSDs, which may be useful in the generation of hypotheses for optimizing their physical activity for preventive care.

The present study aimed to investigate the diurnal profiles of walked steps and locomotive and household activities throughout the day in older adults with MSDs compared with those in age-matched healthy older adults. As chronic pain is the leading complaint of functional limitations in older adults with MSDs, we also examined the effect of the level of chronic pain on the diurnal profiles of locomotive and household activities throughout the day.

## 2. Materials and Methods 

### 2.1. Study Design 

The present study was a cross-sectional survey conducted in three rural areas of Kagoshima prefecture, Japan. Community-dwelling older people (aged 70 years or older) living in Ei-cho, South Kyushu City (total population of Ei-cho area; *n* = 12,388 in 2015), Matsumoto area of the Kagoshima city suburbs (total population of Matsumoto area; *n* = 15,363 in 2015), and Kirishima city (total population of the area; *n* = 125,900 in 2018) were included in this survey. The data of older adults with MSDs were obtained from the clinic in the Ei-cho and Matsumoto areas between February 2015 and November 2018. The older adults with MSDs visited the clinic for treatment of chronic musculoskeletal pain or enhancement of physical function at least one to three times per week. We compared the data of the older adults with MSDs with those of age-matched healthy older adults. The age-matched community-dwelling healthy older adults were recruited from among the participants of the Kirishima Health Welfare Festival 2018 and 2019.

Informed consent was obtained from all participants prior to their entry into this survey. This work was approved by the Ethical Committee of Epidemiological Studies of Kagoshima University (Ref. No.180050).

### 2.2. Participants

The subject enrollment flow chart is shown in Figure 1. The inclusion criteria for older adults with MSDs were age of 70 years or older, independently mobile with or without a walking aid, and community-dwelling (not institutionalized). The exclusion criteria were: orthopedic surgery within the past year, acute pain and history of dementia, neurological disease, cardiovascular disease, or severe comorbid illness. Following screening for inclusion and exclusion criteria, as well as missing data for active monitoring, 32 participants were excluded. Finally, 71 older adults with MSDs provided full activity data and were analyzed with regard to physical activities and physical functions. The older adults with MSDs were almost independent with respect to activities of daily living, with the exception of going up and down stairs.

Age-matched community-dwelling older adults were selected as healthy controls using the following inclusion criteria: age of 70 years or older; independently mobile; no chronic pain, including low back pain and lower extremities pain, at present; no visits to clinics or hospital for treatment of chronic pain within the past year; and no history of orthopedic surgery. Similar to older adults with MSDs, the exclusion criteria were history of dementia, neurological disease, cardiovascular disease and severe comorbid illness. Generally, there is high rate of geriatric syndromes including chronic pain with aging. Therefore, the data of 25 older adults were used as age-matched healthy controls after collecting the data from 70 participants. The activities of daily living of all the control older adults were independent.

### 2.3. Data Collection and Assessment of Physical Functions

Baseline data were collected through a questionnaire that included question on age, sex, body weight, height and chronic pain level. In this study, chronic pain was defined when low back pain and lower extremity pain (e.g., knee pain or hip pain) endured more than 3 months. The degree of the chronic pain was quantified using a numeric rating scale (NRS) and categorized as mild chronic pain (1–3), moderate chronic pain (4–6) and severe chronic pain (7–10).

The timed up and go (TUG) test and the 30 s chair-standing test (CS-30) were performed to assess the physical functions of the participants. The TUG test was performed according to previous reports [14]; the TUG test was conducted twice, and the better value of the two tests was selected as the representative. The participants were asked to sit and stand on a chair with a seat height of 40 cm, as quickly and safely as possible over 30 s periods. The total number of completed chair stands within 30 s was counted and recorded. The CS-30 was conducted once, considering the fatigue of the participants. All measurements were evaluated by trained physical therapists.

### 2.4. Assessment of Physical Activity

The participant daily physical activity was recorded using an Omron activity monitor (Active style Pro, HJA-750C, Omron Healthcare Co., Ltd., Kyoto, Japan). This device was positioned at each participant’s waist. This device calculated the integral of the absolute value of the three signals axes using an accelerated signal over a 10 s time interval [7,12]. After the synthetic acceleration was filtered, it was categorized into either locomotive or lifestyle activity using the ratio of unfiltered-to-filtered synthetic acceleration [7]. The original Omron algorithm can accurately classify locomotive and household activities using the ratios of unfiltered to filtered total acceleration cut off value [12]. Therefore, this device can distinguish not only the intensity of an activity, but also the intensity of basic daily activities, and can classify locomotive and household activities [9,11,12]. Moreover, this device can record the number of steps in daily locomotive activity. In the current study, we analyzed the data on steps and vigorous intensity of locomotive and household activities by MET-hours per week (MET-h/week). The two physical activity intensities were recorded as low (1.0–2.9 METs), moderate (3.0–5.9 METs) and high (6.0 METs or more). The participants were instructed to wear the activity monitor for 7 consecutive days.

### 2.5. Data Analysis

The following data were defined to obtain valid and reliable data on usual daily physical activity: (1) the participants had to wear the device for more than 8 h per day, and (2) the participants had to wear the device for 7 days (from Monday to Sunday). The data collection period was from when the participants got up in the morning until they fell asleep at night. The older adults with MSDs and the older adult controls wore the Omron activity monitor for 11.6 ± 2.6 and 11.6 ± 1.8 (mean ± SD) hours per day, respectively. Data were stored on this device, and the stored activity data were downloaded using the associated software that generated the daily step counts and the locomotive and household activity intensities. Subsequently, the steps per day and steps per hour were calculated, and the steps per hour were analyzed to investigate the diurnal profiles of walking physical activity.

Using the physical activity intensities, we categorized physical activity as locomotive and household activities, and we analyzed the vigorous intensity of locomotive and household activities (MET-h)/week) of moderate and high intensities (≥3 METs). The average locomotive and household activity intensities per hour (MET-h) were calculated as the diurnal profiles of locomotive and household activity per day. The hourly steps and locomotive and household activities throughout the day are illustrated as diurnal profiles of physical activity (mean and 95% confidence interval). To explore whether the physical activities per day (steps and locomotive and household activity intensities) were influenced by chronic pain in older adults with MSDs, we evaluated the chronic pain levels and illustrated the diurnal profiles of physical activity in older adults with mild and severe chronic pain. As the physical activity between 11 p.m. and 5 a.m. was only recorded in a few instances, the graphs of the diurnal profiles cover the period between 6 a.m. and 10 a.m.

The collected data were checked using the Shapiro–Wilk test to determine if they were normally distributed. Subsequently, comparisons between the groups were performed using the Student’s t-test or Mann–Whitney U test. A two-factor repeated measures analysis of variance (groups × time) was used for comparison within- and between-group differences. Statistical analyses were performed using SPSS version 26 (Chicago, IL, USA), and values of *p* < 0.05 were considered statistically significant.

## 3. Results

### 3.1. Participant Characteristics 

The characteristics of all participants are presented in Table 1. There were no significant differences in age, sex, and body mass index between the older adults with MSDs and the control older adults. The older adults with MSDs had two or more orthopedic diseases. All older adults with MSDs had chronic pain, and 18.3% had severe chronic pain.

### 3.2. Physical Activity and Diurnal Profiles Per Day

The physical function and locomotive and household activities per week were compared between older adults with MSDs and control older adults (Table 2). All physical functions and step parameters in older adults with MSDs were significantly lower than those in older adult controls. The number of steps per day and the number of steps per hour in older adults with MSDs were approximately 50% lower than those in older adult controls. The activity monitor-determined locomotive activity (MET h/week) in older adults with MSDs was approximately 20% worse than that in the older adult controls. However, the household activity (MET h/week) in older adults with MSDs was similar to that in older adult controls, showing that the household activity in older adults with MSDs was maintained, even if the physical functions and locomotive activity declined, compared with that in age-matched healthy older adults.

Over the course of the day, the diurnal profiles of locomotive and household activities in older adults showed two peaks of physical activity in the morning and afternoon, and one low point of physical activity around 12 p.m. or 1 p.m (Figure 2). Furthermore, the diurnal profiles of steps and activity monitor-determined locomotive activity were significantly different between group and time interactions (*p* < 0.001), and showed different physical activity patterns throughout the day. In the older adults with MSDs, the number of steps per hour peaked at 9 a.m. (mean: 247; 95% CI: 192–301) and 3 p.m. (mean: 240; 95% CI: 183–297), and locomotive intensity activity per hour peaked around 9 a.m. (mean: 0.26; 95% CI: 0.15–0.37) and 3 p.m. (mean: 0.22; 95% CI: 0.12–0.32). In contrast, the number of steps per hour of the older adult controls peaked at 11 a.m. (mean: 504; 95% CI: 331–676) and 5 p.m. (mean: 621; 95% CI: 244–999), and their locomotive activity peaked around 11 a.m. (mean: 1.11; 95% CI: 0.43–1.78) and 5 p.m. (mean: 1.80; 95% CI: 0.25–3.35). The two peaks of physical activity in older adults with MSDs were considerably lower than those of the older adult controls. These profiles show that older adults with MSDs have reduced locomotive activity throughout the day compared with age-matched healthy older adult controls.

The household activity of older adults with MSDs peaked around 9 a.m. (mean: 1.55; 95% CI: 1.30–1.80) and 4 p.m. (mean: 1.32; 95% CI: 1.09–1.56), while in the older adult controls, household activity (MET hour) peaked around 9 a.m. (mean: 1.58; 95% CI: 1.22–1.94) and 4 p.m. (mean: 1.19; 95% CI: 0.93–1.45). There were no significant differences between group and time interactions in the diurnal profile of household activity (*p* = 0.060), and the analysis of between-subject effects showed no significant differences in steps (*p* = 0.185). The diurnal profile of household activity in the older adults with MSDs was similar to that in the age-matched healthy older adult controls.

### 3.3. Chronic Pain Level and Physical Activity in Older Adults with MSDS 

The differences in locomotive and household activities classified by chronic pain level are shown in the Table 3. The locomotive and household activities were influenced by chronic pain level, and the chronic pain level negatively impacted instrumental activities of daily living. The number of steps per day in older adults with severe pain was significantly lower than that in older adults with mild pain. Therefore, we compared the diurnal profiles of locomotive and household activities (MET hour) per day throughout the day in older adults with mild and severe pain levels (Figure 3). There was no significant difference between the group and time interactions in the diurnal profiles of steps (*p* = 0.458) and locomotive activity (*p* = 0.722), and the analysis of between-subject effects showed no significant differences in steps (*p* = 0.053) and locomotive activity (*p* = 0.052). The diurnal profile of household activity throughout the day showed a significant difference between group and time interactions (*p* = 0.035), suggesting that the diurnal profile of household activity throughout the day was affected by chronic pain level.

## 4. Discussion

The present study investigated the diurnal profiles of steps and locomotive and household activities, which were measured using three axial accelerations, in community-dwelling older adults with MSDs. Mai et al. [6] demonstrated that the pedometer-determined diurnal profiles of physical activity by steps in chronically ill and mobility-limited older adults showed two peaks in the morning and in the afternoon. The diurnal profiles of steps and the locomotive and household activities of older people with MSD were similar in our study to the results of previous reports. The current study showed that phases of higher activity in the morning and afternoon were usually occupied by household activity, and not locomotive activity, in older adults with MSD. Several studies have reported that limited mobility decreases physical activity [5,6]. Furthermore, the present survey showed that physical function and locomotive activity were considerably lower in the older people with MSDs, while the diurnal profile of the steps and locomotive activity were clearly different between the older people with MSDs and age-matched older adults. However, the household activity and diurnal profiles of household activity throughout the day were similar in both groups of older adults. These findings suggest that household activity, but not locomotive activity, was maintained in the daily life of older adults with MSDs, although their physical function and locomotive activity were declined considerably throughout the day.

A wide range of epidemiological studies have examined physical activity in elderly people [15,16,17]. The normative data of recent reviews indicate that healthy older adults take 2000–9000 steps/day on average, and special populations, such as those living with disability and/or chronic illness, average 1200–8800 steps/day [4]. In addition, Tudor-Locke et al. [18] conducted a literature review and suggested the use of pedometer-determined habitual physical activity in adults (e.g., sedentary: <5000 steps/day, low active: 5000–7499 steps/day, physically active; >7500 steps/day). According to these categories, our results showed that mean steps/day of the older adults with MSDs were categorized as sedentary, while those of the age-matched healthy older adults were categorized as low active. Physical activity declines with increasing age, and is generally cause by frailty or sarcopenia [19]. Our results showed that chronic pain was associated with a low diurnal profile of steps and locomotive activity. Chronic musculoskeletal pain in community-dwelling older adults can be associated with frailty and pre-frailty, and geriatric syndromes, including chronic pain, cause deterioration in the quality of life in community-dwelling older adults [20,21]. However, promoting an increase in locomotive activity is beneficial for reducing pain and improving the quality of life in the adult population with MSDs [3].

In contrast, the household activity of the older adults with MSDs did not decline compared with that of age-matched healthy older adults, even if they had chronic pain and reduce physical function. Therefore, community-dwelling older people with MSDs may be active in regard to household activities in daily living, while also controlling chronic pain. Such activities may constitute an integral part of the daily routines, which may improve health, well-being, and sleep quality in older adults with sedentary or low activity levels [6]. However, the older adults with severe chronic pain showed a considerable decline in the diurnal profile of household activity. Therefore, pain management may be an important strategy to help older people with MSDs to maintain daily household activity.

A decline in physical activity is associated with poor physical performance [22,23]. Walking is the most common physical activity and has been shown to be associated with various benefits for older persons [24]. A physical activity of more than 3.0 METs is correlated with increased bone mineral density in men and women aged 70 years old [25]. In addition, high-intensity activity is related to lower levels of pain in patients with spinal cord injury [26]. Furthermore, in the older adults with MSDs, household activity of more than 3.0 METs was considerably higher than locomotive activity of more than 3.0 METs. Therefore, it is often important for older people to maintain household activities throughout the day, both for the promotion of physical activity and the increase in the locomotive activity.

In addition, Mansi et al. [3] established that participation in walking exercise for 3 or 4 days per week was effective for increasing physical activity and reducing pain in people with MSDs. Indeed, several studies have supported the use of walking-based interventions to encourage people with a range of MSDs to assume more physically active roles in their management [3,27]. In an animal study, running exercises performed 3 or 5 days per week reduced neuropathic pain through regulation of the endogenous opioid system and neurotrophic factor in the brain stem or spinal cord [28]. Physical function is related to moving and performing daily activities [22], and decreased muscle strength is associated with decreased physical function [24]. Therefore, it may also be important for older adults with MSDs to increase walking activities throughout the day. As the diurnal profiles of locomotive activity of older adults with MSDs were considerably lower in the morning and afternoon, physical activity in the morning or afternoon should be indicated for preventive care or outpatient rehabilitation services.

There were several limitations to the present study. First, this study assessed physical activity using an Omron activity monitor consisting of triaxial acceleration sensors. An accelerometer can provide additional data with regard to the time spent in various intensities of physical activity and inactivity as well as step count data [4]. However, the acceleration sensors may be less sensitive to the movement of the center of gravity, such as that in very slow walking [29]. In addition, measurement data acquired by an accelerometer may vary depending on the measurement system of each device. As the Omron activity monitor calculates the integral of the absolute value of the axes of the three signals using an accelerated signal over a 10 s recorded time interval [7,12], it may be calculated with a slightly higher activity intensity. Therefore, in this study, we compared the data of older adults with MSDs and age-matched healthy older adults using same triaxial accelerometer. Second, we did not examine internal diseases, such as diabetes or chronic obstructive pulmonary diseases, in the participants. These chronic internal diseases may have an influence on the physical activity of the day. Therefore, we excluded those with severe comorbid illnesses from this study. Third, we did not examine medication, including painkillers or antihypertensive agents, in the questionnaire. Taking painkillers might influence the daily physical activity of older adults. However, pharmacological treatment for chronic pain is limited, with no more than 40–60% of patients obtaining partial relief of their pain [30]. Fourth, the diurnal profiles of physical activity may be affected by several factors, including climatic conditions, socioeconomic status, education and gender. The activity range of older adults decreases during the winter season compared with the summer season [23,31]. Furthermore, we did not examine the socioeconomic status and education level of the older adults. Therefore, further studies are need to examine the factors influencing diurnal profiles of physical activity. Despite these limitations, this is the first study to explore the diurnal profiles of the physical activities that were classified into locomotive and household activities in community-dwelling older people with MSDs. Community-dwelling older people with MSDs maintained household activities throughout the day, despite reduced physical functions and locomotive activity. However, severe chronic pain was shown to reduce both household activity and locomotive activities.

## 5. Conclusions

The diurnal profiles of household activity in older people with MSDs were not reduced compared with those in age-matched healthy older people, although their locomotive activity was considerably lower than that of healthy older people. However, an increase in chronic pain was associated with a decline in household activity. It may be an important strategy to prevent the decline of household activity due to chronic pain throughout the day for the promotion of physical activity in older adults with MSDs. Our findings provide a basis for developing strategies for the enhancement and maintenance of physical activity in community-dwelling older people with MSDs.

## Figures and Tables

**Figure 1 ijerph-17-05337-f001:**
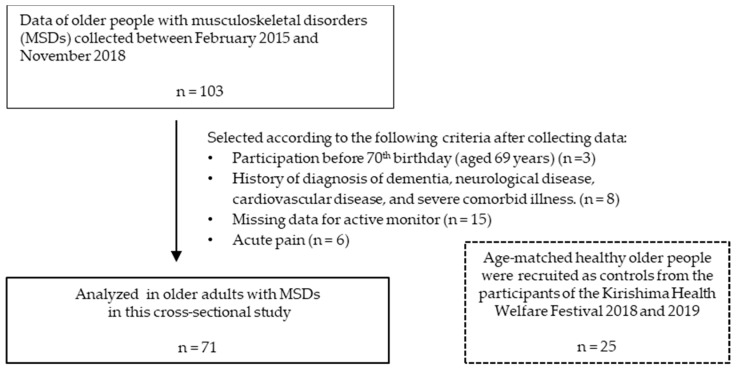
Flow chart of participant selection for this study.

**Figure 2 ijerph-17-05337-f002:**
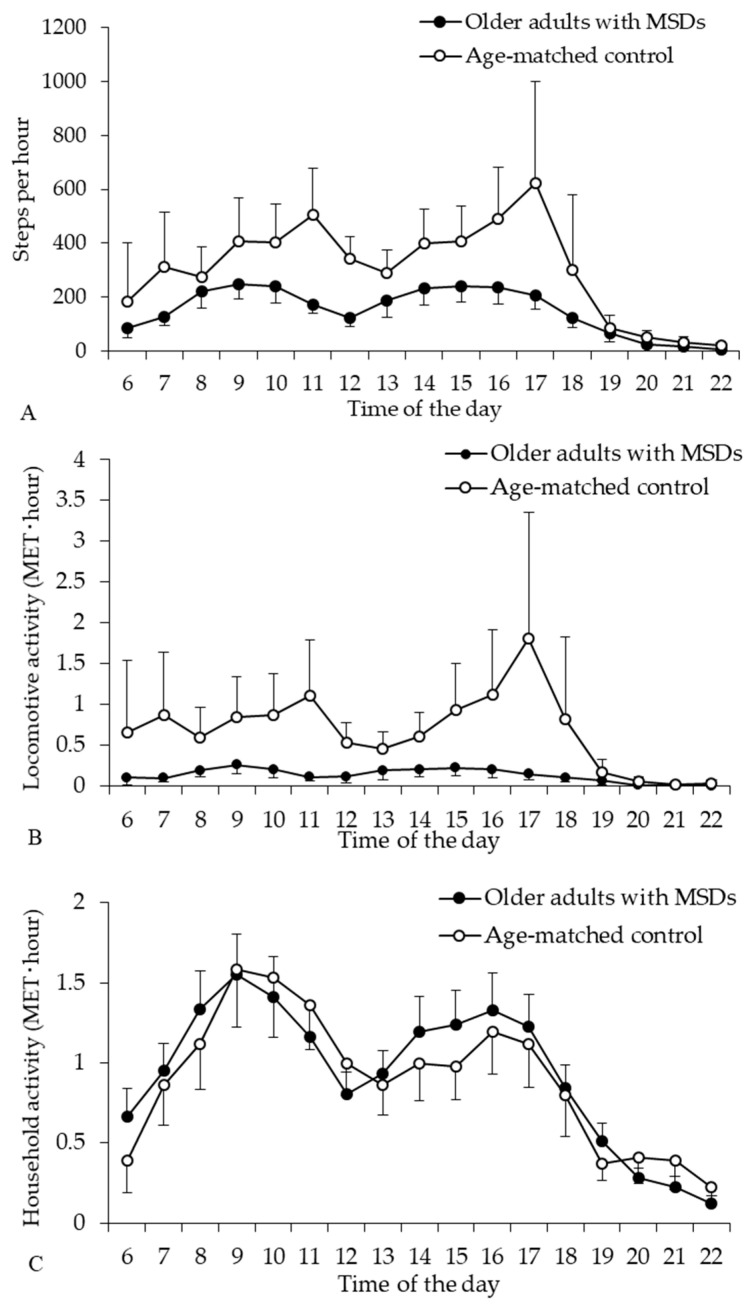
Diurnal profile of steps (**A**) and locomotive (**B**) and household (**C**) activities in older adults with MSDs and age-matched healthy older adults. Mean ± 95% confidence intervals.

**Figure 3 ijerph-17-05337-f003:**
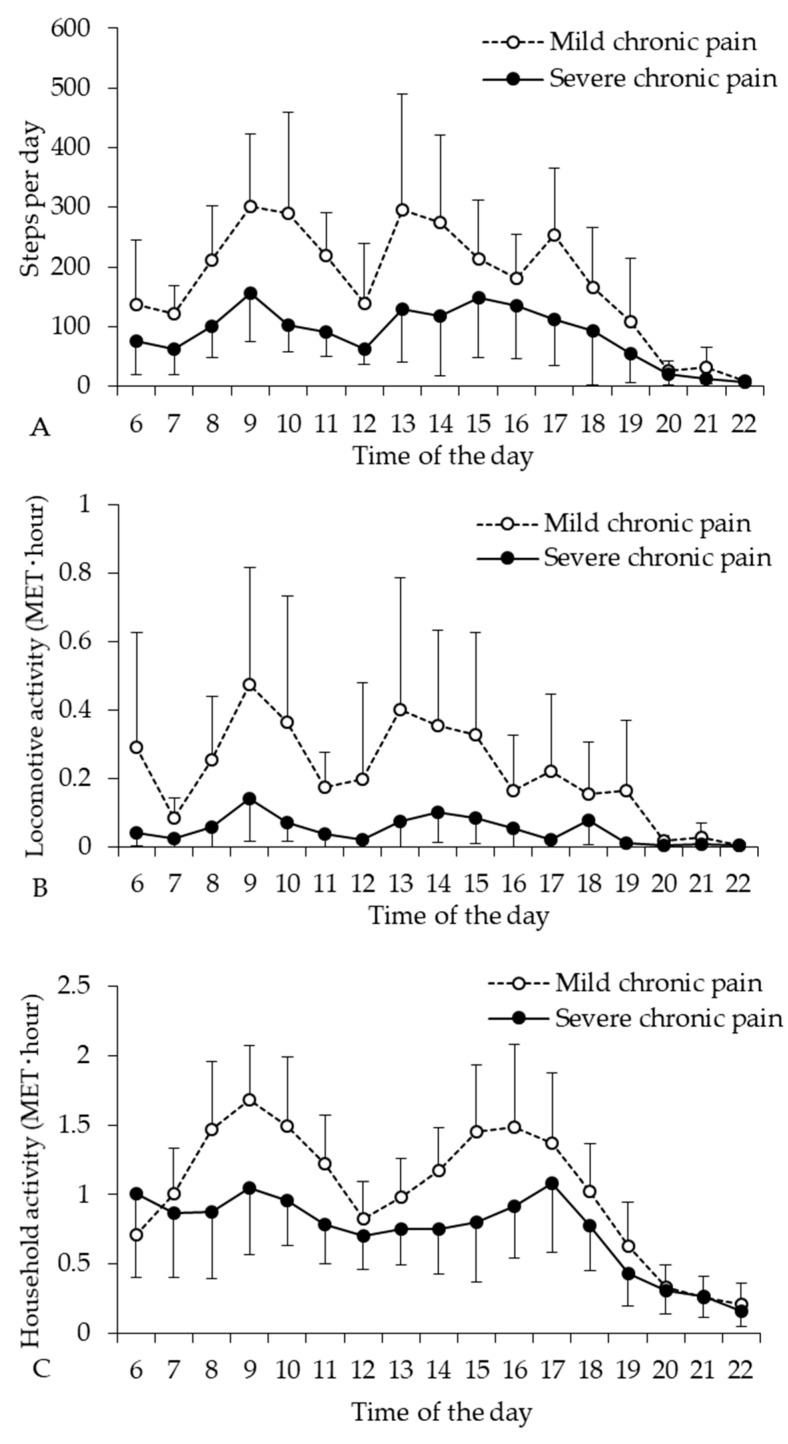
Diurnal profile of steps (**A**) and locomotive (**B**) and household (**C**) activities adjusted by chronic pain level (mild and severe) in older adults with MSDs. Mean ± 95% confidence intervals.

**Table 1 ijerph-17-05337-t001:** Participant characteristics.

	Older Adults with MSDs (*n* = 71)	Control Older Adults (*n* = 25)	*p*-Value
Age (years)	81.1 ± 5.0	79.5 ± 3.5	0.072
Men/women (n)	15/56	10/15	0.064
Body mass index (kg/m^2^)	23.7 ± 3.3	22.5 ± 2.6	0.097
Diagnosis of orthopedic disease for rehabilitation * (n)			
Osteoarthritis of the knee	31	―	
Chronic low back pain	17	―	
Lumbar spondylosis	17	―	
Lumbar spinal stenosis	10	―	
Conservative therapy after fracture (e.g., lumbar vertebrae, patella)	2	―	
After surgery of the lower limbs (e.g., TKA, THA, osteosynthesis)	9	―	
After surgery of lumbar or cervical spine (e.g., PLF, depression, lumber disc herniation)	6	―	
Chronic pain (NRS) (n)			<0.001
None (0)	0	25	
Mild (1–3)	20	0	
Moderate (4–6)	38	0	
Severe (7–10)	13	0	

Mean ± standard deviation (SD). MSDs: musculoskeletal disorders; * participants have more than one type of diagnosis. TKA: total knee arthroplasty; THA: total hip arthroplasty; PLF: posterior lumbar fusion; NRS: numerical rating scale.

**Table 2 ijerph-17-05337-t002:** Physical functions and physical activity parameters in older adults with MSDs and controls.

	Older Adults with MSDs (*n* = 71)	Control Older Adults (*n* = 25)	*p*-Value
Physical functions			
Timed Up and Go test (second)	12.7 ± 4.5	6.6 ± 1.0	<0.001
CS-30 test (repetitions)	11.8 ± 3.9	17.5 ± 5.2	<0.001
Steps parameters (steps)			
Steps per day	2563.8 ±2063.7	5137.6 ± 2844.2	<0.001
Steps per hour	150.9 ± 121.3	302.2 ± 167.3	<0.001
Physical activities (MET-h/week) *			
Locomotive activity	2.3 ± 3.1	11.5 ± 10.9	<0.001
Household activity	15.8 ± 8.9	15.1 ± 5.5	0.667

Mean ± SD. CS-30: 30 s chair-standing test. * Energy consumption for locomotive and household activities correspond to physical activity more than 3.0 METs.

**Table 3 ijerph-17-05337-t003:** Chronic pain level and physical activity in older adults with MSDs.

	Steps Per Day (Steps)	Locomotive Activity (MET-h/Week)	Household Activity (MET-h/Week)
Chronic pain (*n* = 71)			
Mild (*n* = 20)	2987.2 ± 2152.6	3.7 ± 4.7	17.9 ± 9.0
Moderate (*n* = 38)	2717.5 ± 2150.4	2.1 ± 2.3	16.5 ± 9.2
Severe (*n* = 13)	1490.1 ± 1268.7	0.8 ± 0.8	12.5 ± 7.5
*p*-value (Mild vs. Severe)	0.032	0.065	0.147

Mean ± SD. MET: metabolic equivalent of task

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
