# Peer review of "Diurnal Profiles of Locomotive and Household Activities Using an Accelerometer in Community-Dwelling Older Adults with Musculoskeletal Disorders: A Cross-Sectional Survey"

_ijerph, 2020, doi:10.3390/ijerph17155337_

Round 1

Reviewer 1 Report

Thank you for sending me this manuscript to review. It compares the diurnal activity profiles of older people with and without MSK disorders. The study design is a straightforward cross-sectional comparison of data between the two groups. The analysis is done well and the conclusions support the data. Graphs and tables are clear. The findings are interesting and provide some clear implications regarding PA in older people. Most of the limitations are outlined. 

I have a few comments:

  1. My main concern is that the effect of season is not accounted for. The periods of data collection are very long, and it is highly likely that locomotor activity would be reduced in more extreme temperatures of hot or cold. The season of collection should be described in both groups as this is a potential confounding variable. The impact of this should be discussed in the discussion. 
  2. Likewise other potential differences between the healthy and MSK group should be noted - were socioeconomic status and education collected? The discussion section should consider alternative explanations for the differences seen between the two groups. 
  3. I am slightly surprised, given the nature of the study and previous work in this area, that the authors did not state clear prior hypotheses about what they would would find. 
  4. Somewhere in the paper it needs to be stated where the activity monitor is worn on a person.  Further detail on the validity and reliability of the Omron monitor is needed - currently only references are briefly given. Was this validated in older populations (who may move more slowly) and in those with MSK disease, and what were the measurement property statistics for validity and reliability of activity classification?  
  5. More elaboration is needed in the introduction regarding the findings of the other studies on diurnal profiles - it states only that they were carried out. Further detail would contextualise the present study better.
  6.  This sentence needs rephrasing "The older adults with MSDs visited the clinic for treatment of chronic musculoskeletal pain or enhancement of physical function at least one to three times per week, and their data were compared with those of the older adults with MSDs with that of age-matched healthy older adults."
  7. There needs to be further explanation given for the choice of sample size for healthy controls. 
  8. Between the text and flow chart it is unclear whether participants were excluded at screening level or after collecting data. This needs to be clarified. Also, no exclusion reasons are given in text or the flow chart for healthy controls. This should be added. 
  9. Did the authors consider looking at the impact of different MSK diseases upon activity profiles?
  10. Line 189 - presumably 12AM should be 12PM

Author Response

1. My main concern is that the effect of season is not accounted for. The periods of data collection are very long, and it is highly likely that locomotor activity would be reduced in more extreme temperatures of hot or cold. The season of collection should be described in both groups as this is a potential confounding variable. The impact of this should be discussed in the discussion.

Answer

We agree your opinion. As Kagoshima prefecture is southern Japan, there is relatively little snow in winter season. However, we did not analysis seasonal difference. Therefore, we add the sentence in the limitation in the discussion section.

ijerph-857974 original corrected revision file

Line 324-

Fourth, diurnal profiles of physical activity may be affected by several factors including climatic conditions, socioeconomic status, education, and gender. The activity range of older adults decreases during the winter season compared with the summer season [23, 31]. Furthermore, we did not examine the socioeconomic status and education level of the older adults. Therefore, further studies are need to examine the factors influencing diurnal profiles of physical activity.

2. Likewise other potential differences between the healthy and MSK group should be noted - were socioeconomic status and education collected? The discussion section should consider alternative explanations for the differences seen between the two groups.

Answer

We agree your opinion. In this survey, we did not examine the socioeconomic status and education level of the older adults. Therefore, we add the sentence in the discussion section.

ijerph-857974 original corrected revision file

Line324-

Fourth, diurnal profiles of physical activity may be affected by several factors including climatic conditions, socioeconomic status, education, and gender. The activity range of older adults decreases during the winter season compared with the summer season [23, 31]. Furthermore, we did not examine the socioeconomic status and education level of the older adults. Therefore, further studies are need to examine the factors influencing diurnal profiles of physical activity.

3. I am slightly surprised, given the nature of the study and previous work in this area, that the authors did not state clear prior hypotheses about what they would would find.

Answer

We are sorry. This comment is hard to be understood for us, but we want to answer as far as we understood it.

In previous study, physical activity questionnaire could be feasible approach to activity energy expenditure estimation in large population, but it is unclear whether or not any physical activity questionnaire is valid for estimating activity energy expenditure. Recently, we can evaluate it with an accelerometer quantitatively. Therefore, we examined the locomotive and household activity intensities by metabolic equivalent task-hours per week (MET-h/week) quantitatively. We add the sentence in the introduction section. And we omitted the sentence.

ijerph-857974 original corrected revision file

Line 46-

Mai et al [6] reported that the high activity phases throughout the day were mostly due to basic and instrumental activities of daily living, such as housework. However, they did not actually investigate the diurnal profiles that categorized locomotive or household activities in older adults. Therefore, it is unclear whether the older adults performed housework activities in high activity phases throughout the day.

Line258-

We omitted the following sentence.

Furthermore, it was assumed that the high activity phases throughout the day were mostly due to basic and instrumental activities of daily living, such as housework [6].

4. Somewhere in the paper it needs to be stated where the activity monitor is worn on a person. Further detail on the validity and reliability of the Omron monitor is needed - currently only references are briefly given. Was this validated in older populations (who may move more slowly) and in those with MSK disease, and what were the measurement property statistics for validity and reliability of activity classification?

Answer

We agree your opinion. Therefore, we included the additional three references for the validity and reliability of the Omron monitor. A literature number was thereby changed. Please check it. Then we add the sentence in the Introduction and Materials and Methods sections.

ijerph-857974 original corrected revision file

Line 57-

Accelerometers are non-invasive tools for measuring physical activity and activity energy expenditures, with the potential to measure locomotive as well as household activities [8-10].

Line 66

Many investigators have assessed the validity and reliability of the Omron activity monitor [9,11,12].

Line 133

This device was positioned at each participant’s waist.

Line 373-

We included the following additional three references.

  1. Trost SG, McIver KL, Pate RR. Conducting accelerometer-based activity assessments in field-based research. Med Sci Sports Exerc. 2005;37: S531-543.
  2. Ohkawara K, Oshima Y, Hikihara Y, Ishikawa-Takata K, Tabata I, Tanaka S. Real-time estimation of daily physical activity intensity by a triaxial accelerometer and a gravity-removal classification algorithm. Br J Nutr. 2011; 105:1681-1691.
  3. Chen KY, Bassett DR Jr. The technology of accelerometry-based activity monitors: current and future. Med Sci Sports Exerc. 2005;37: S490-500.

5. More elaboration is needed in the introduction regarding the findings of the other studies on diurnal profiles - it states only that they were carried out. Further detail would contextualise the present study better.

Answer

We add the sentence in the introduction regarding the findings of the other studies.

ijerph-857974 original corrected revision file

Line 47-

Mai et al [6] reported that the high activity phases throughout the day were mostly due to basic and instrumental activities of daily living, such as housework. However, they did not actually investigate the diurnal profiles that categorized locomotive or household activities in older adults. Therefore, it is unclear whether the older adults performed housework activities in high activity phases throughout the day.

6. This sentence needs rephrasing "The older adults with MSDs visited the clinic for treatment of chronic musculoskeletal pain or enhancement of physical function at least one to three times per week, and their data were compared with those of the older adults with MSDs with that of age-matched healthy older adults."

Answer

We corrected these sentences.

ijerph-857974 original corrected revision file

Line 87-

The older adults with MSDs visited the clinic for treatment of chronic musculoskeletal pain or enhancement of physical function at least one to three times per week, and their data were compared with those of the older adults with MSDs with that of age-matched healthy older adults.

→→The older adults with MSDs visited the clinic for treatment of chronic musculoskeletal pain or enhancement of physical function at least one to three times per week. We compared with the data of the older adults with MSDs with that of age-matched healthy older adults.

7. There needs to be further explanation given for the choice of sample size for healthy controls.

Answer

We agreed your opinion. We corrected and add the sentence in the Materials and methods section.

ijerph-857974 original corrected revision file

Line 117-

Generally, there is high rate of geriatric syndromes including chronic pain with aging.

Line 118

After 70 participants were screened, the data of 25 older adults were used as age-matched healthy controls.

→→Therefore, the data of 25 older adults were used as age-matched healthy controls after collecting the data from 70 participants.

8. Between the text and flow chart it is unclear whether participants were excluded at screening level or after collecting data. This needs to be clarified. Also, no exclusion reasons are given in text or the flow chart for healthy controls. This should be added.

Answer

We agreed your opinion. We corrected the sentence in Figure 1 and text. Exclusion criteria in the control group is already included in the text (Line 112~). Please confirm it.

Figure 1

Selected according to the following criteria:

→→ Selected according to the following criteria after collecting data:

Line 118

Therefore, the data of 25 older adults were used as age-matched healthy controls after collecting the data from 70 participants.

Line 108~

Age-matched community-dwelling older adults were selected as healthy controls using the following inclusion criteria: Age of 70 years or older; independently mobile; no chronic pain, including low back pain and lower extremities pain, at present; no visits to clinics or hospital for treatment of chronic pain within the past year; and no history of orthopedic surgery. Similar to older adults with MSDs, the exclusion criteria were history of dementia, neurological disease, cardiovascular disease, and severe comorbid illness.

9. Did the authors consider looking at the impact of different MSK diseases upon activity profiles?

Answer

We did not examine the impact of different MSDs on activity profiles. Some adults with MSDs have several geriatric symptoms. Further, participants have more than one type of diagnosis. Therefore, it may be difficult to examine by classify disease.

10. Line 189 - presumably 12AM should be 12PM

Answer

Thank you for indication. We corrected it.

Line 202

12AM

→→ 12PM

Reviewer 2 Report

In my opinion, this is a relevant study with some good insights, but also with some predictable results and conclusions.For instance, it is quite obvious that the more severe the chronic pain, the greater the reduction in both household activity and locomotive activities.I would suggest careful editing to reduce the attention towards such common knowledge and highlight what's new.

Indeed, previous studies have shown that promoting an increase in locomotive activity is beneficial for reducing pain and improving the quality of life in the adult population with MSDs.But the data and analysis in this article do not provide evidence that household activity can promote or increase the locomotive activity. There was no significant difference in household activity between the older adults with MSDs and the age-matched healthy controls, which was more likely due to the necessity of survival. Therefore, it remains to be seen whether maintaining household activities throughout the day is an effective strategy.

In addition, I wonder if the authors noticed the fact that the ratio of males to females in the experimental group and the control group is quite different. The influence of gender may be important, but there is no discussion in the article. Perhaps this is limited by the availability of samples, but I think relevant discussions should at least be mentioned.

Author Response

Reviewer #2:

Comments and Suggestions for Authors

1. In my opinion, this is a relevant study with some good insights, but also with some predictable results and conclusions.For instance, it is quite obvious that the more severe the chronic pain, the greater the reduction in both household activity and locomotive activities.I would suggest careful editing to reduce the attention towards such common knowledge and highlight what's new.

Answer

Thank you for your comments. However, some studies did not actually investigate the diurnal profiles that categorized locomotive or household activities in older adults. Therefore, it is unclear whether the older adults performed housework activities in high activity phases throughout the day. this is the first study to explore the diurnal profiles of the physical activities that were classified into locomotive and household activities in community-dwelling older people with MSDs. Therefore, we add the introduction section

Line 47-

Mai et al [6] reported that the high activity phases throughout the day were mostly due to basic and instrumental activities of daily living, such as housework. However, they did not actually investigate the diurnal profiles that categorized locomotive or household activities in older adults. Therefore, it is unclear whether the older adults performed housework activities in high activity phases throughout the day.

2. Indeed, previous studies have shown that promoting an increase in locomotive activity is beneficial for reducing pain and improving the quality of life in the adult population with MSDs.But the data and analysis in this article do not provide evidence that household activity can promote or increase the locomotive activity. There was no significant difference in household activity between the older adults with MSDs and the age-matched healthy controls, which was more likely due to the necessity of survival. Therefore, it remains to be seen whether maintaining household activities throughout the day is an effective strategy

Answer

We think that it is important to increase the both locomotive and household activities. A physical activity of more than 3.0 METs is correlated with increased bone mineral density in men and women aged 70 years old. Therefore, Household activity more than 3.0 METs may be promote physical health for older adults with MSDs. It is important that we increase activity while living. Our results suggest that the household activity was preserved in the life of the day in the MSDs group, although the physical function and locomotive activity were greatly decreased. In addition, community-dwelling older people in rural areas may have to perform housework of daily living independently, which was more likely due to the necessity of survival (like your comment) . Therefore, physical therapy improves the daily physical activity of community-dwelling elderly people with MSDs by regular rehabilitation interventions such as chronic pain management, increased leg muscle strength, improved balance ability, and maintenance of instrumental activities of daily living. Taken together, we believe that the maintenance of household activity may be important strategies for the promotion of physical activity in the older people with MSDs.

3. In addition, I wonder if the authors noticed the fact that the ratio of males to females in the experimental group and the control group is quite different. The influence of gender may be important, but there is no discussion in the article. Perhaps this is limited by the availability of samples, but I think relevant discussions should at least be mentioned.

Answer

In this study, there is no significant difference in gender between groups. household and locomotive activities is affected by gender. We analysis the difference in gender in older adults with MSDs. According our analysis, in older adults with MSDs, locomotive activity was high in men (3.4± 4.2 vs 1.9 ± 2.6; Men vs Women), and household activity was high in women (12.8 ± 9.6 vs 16.3 ± 8.7; Men vs Women). The household activity of women was significantly higher than that of men (p < 0.05). Step counts/ day was not significant difference between groups (2659.1 ± 2405.8 vs 2457.5 ± 1985.7; Men vs Women) In healthy older adults, it was a similar tendency. However, locomotive activity and step counts of men was significantly higher than that of women (p < 0.05; 20.2 ± 17.0 vs 8.6 ± 6.8 in locomotive activity; 7414.4 ± 4425.2 vs 4758.4 ± 2238.5 in steps). Household activity and was not significant difference in gender (15.7 ± 6.5 vs 21.4 ± 8.4; Men vs Women). However, this study mainly aimed that the diurnal profiles of steps and locomotive and household activities throughout the day in older adults with MSDs compared with those in age-matched healthy older adults. Therefore, we add the sentences in discussion section.

Line324-

Fourth, diurnal profiles of physical activity may be affected by several factors including climatic conditions, socioeconomic status, education, and gender. The activity range of older adults decreases during the winter season compared with the summer season [23, 31]. Furthermore, we did not examine the socioeconomic status and education level of the older adults. Therefore, further studies are need to examine the factors influencing diurnal profiles of physical activity.